# OpenReview forum: "Aligned LLMs Are Not Aligned Browser Agents"
_ICLR.cc/2025/Conference — ICLR 2025 Poster_

### Official Review · Reviewer_DmgK · 2024-10-20

**Soundness:** 3
**Presentation:** 3
**Contribution:** 3
**Rating:** 8
**Confidence:** 4

**Summary:**

This paper introduces and publicly releases Browser Agent Red teaming Toolkit (BrowserART), a test suite for evaluating the safety of browser-based LLM agents. The authors compile 100 browser-related harmful behaviors + 40 synthetic websites to test if safety-trained LLMs maintain their guardrails when used as browser agents. Their experiments reveal a significant drop in safety alignment between LLMs and their corresponding browser agents, with browser agents more likely to execute harmful behaviors. The paper also demonstrates that existing techniques for jailbreaking LLMs transfer effectively to browser agents.

**Strengths:**

- Addresses an important, emerging, and under-explored area of AI safety: LLM-based browser agents.
- Promotes collaboration on improving LLM agent safety by publicly releasing Browser ART with 100 harmful behaviors and 40 synthetic websites.
- The paper is well-structured and clearly written.
- Clearly demonstrates that standard safety training of LLMs does not necessarily transfer to browser agent applications.
- A range of SOTA LLMs are evaluated including models from the OpenAI GPT, Anthropic Claude, Meta Llama, and Google Gemini families.

**Weaknesses:**

No substantive weaknesses.

While the paper identifies the gap in alignment when LLMs are used as browser agents, it does not investigate the underlying reasons. More analysis on why browser agents are more vulnerable could provide valuable insights.

**Questions:**

- Have you investigated the alignment gap in other kinds of AI agents beyond browser agents?
- What specific changes to LLM safety training do you believe could help address the alignment gap observed in browser agents?

---

> ### Author Response · Authors · 2024-11-21
> **Response**
>
> Thank you for your encouraging feedback and positive endorsement of our work!
>
>
> > Have you investigated the alignment gap in other kinds of AI agents beyond browser agents?
>
> We have not because we are not entirely convinced other types of agents are worth this type of red teaming. The reason is stated in the introduction and we summarize here: compared to calling specific APIs or using code interpreters, browser agents are much more generic and universal because the tool itself (i.e. the browser) can be used in a general way. Perhaps the most similar direction is GUI agents; however, at least when this work is created, GUI use is still a much more difficult problem compared to browser use and we cannot confidently claim that GUI agents are capable enough to complete even regular tasks, let alone dangerous ones. Both browser and GUI agents are under active, rapid development; capable browser agents just so happen to have been developed slightly earlier than GUI agents, so this work focuses on the former. We also wanted to release this work as soon as possible to highlight the importance of expanding agent safety before autonomous browser agents are actually launched onto the Internet. The good news is that since the computer-use Sonnet-3.5 was released, GUI-related agent scaffoldings and investigation will follow.
>
> > What specific changes to LLM safety training do you believe could help address the alignment gap observed in browser agents?
>
> As per our hypotheses of this observed alignment gap (Section 5: Why Agents Are Less Safety-Aligned), we believe this unsafe behavior of browser agents can be mitigated by introducing safety training data related to agentic browser use-cases for LLMs.
> The training data for agentic use is trajectory data – a list of tuples of environment state, thought, and the desired action to take. For a browser agent, the state is a browser event. A safety-related data sample for browser agents in this case consists of a trajectory of the previously-taken actions all the way until the step where the agent must reject due to the risk of the following action to cause harm. By having a refusal trajectory, the agent learns to predict the impact of its action to the environment (i.e. the Internet) internally before taking an action.
>
> > While the paper identifies the gap in alignment when LLMs are used as browser agents, it does not investigate the underlying reasons.
>
> This is indeed a missing part in this work and the reason is two-fold. First, we want to just entirely focus on presenting this interesting finding and showcase that this type of jailbreak can be done in an extremely simple way. Sometimes, one does not need to create jailbreaks at all and **the process of agent development is itself an unintentional jailbreak.** We have spent efforts looking into many hypotheses for the reason behind the alignment gap but could not confidently claim one. Besides, it is hard to dive deep and do systematical ablation study in a short conference paper while focusing on presenting the red teaming results. Second, it is likely there are confounding reasons instead of one obvious “bug” behind this result. In hindsight, the process of understanding the adversarial vulnerability in conventional CNNs also took several years since it was first discovered in 2013. The most notable work in explaining adversarial vulnerability is [1], which is about 6 years after the classic panda-gibbon example. Therefore, acknowledging the issue and solving the issue are likely to take more time than one publication. However, luckily, LLM research evolves faster and we are working on follow-ups as well.
>
> [1] Ilyas, Andrew, et al. "Adversarial examples are not bugs, they are features." Advances in neural information processing systems 32 (2019).

---

> > ### Comment · Reviewer_DmgK · 2024-11-27
> >
> > Thank you for the additional details. I will maintain my score as it is already high.

---

### Official Review · Reviewer_cbCU · 2024-11-03

**Soundness:** 2
**Presentation:** 3
**Contribution:** 2
**Rating:** 6
**Confidence:** 4

**Summary:**

The paper introduces a benchmark that consists of 100 harmful behaviors for red-teaming web agents. The experiments on two types of web agents demonstrate that the web agents are not well aligned even though their underlying model is aligned. Also, existing jailbreaking attacks can transfer well to the agent setting.

**Strengths:**

1. I believe red-teaming web agents is an important direction. Although lots of concurrent works have already been released, the authors are the first, to the best of my knowledge, to propose such a benchmark by the deadline of paper submission. Therefore, I believe this is one of the earliest works in this direction.

**Weaknesses:**

1. The paper severely lacks a comprehensive literature review and comparison against existing red-teaming methods for agents. There has been a bunch of related work in this domain, such as [1]. At least some discussions should be provided to demonstrate the difference and novelty of the proposed toolkit.

2. The benchmark is relatively too small, contains only 100 examples, and from my understanding, is not scalable. Also, the hierarchy of risk categories is not well-supported. There have been a lot of safety regulations or risk categories proposed by companies, governments, or researchers, such as [2]. However, the hierarchy here is only a small subset. As a result, the benchmark is not comprehensive.

3. The red-teaming method introduced in the paper lacks novelty. Most examples are manually crafted or transformed from existing examples, and there is no automated way of doing that (or at least not mentioned by the authors). Is it possible to formalize an algorithm to automatically perform red-teaming? Are there any specific characteristics of web agents that should be considered when red-teaming web agents instead of the LLM? Have the authors made efforts to make the existing jailbreaking attacks more effective against agents? Simply designing 100 examples for red-teaming does not bear enough novelty.

[1] Tian, Yu, et al. "Evil geniuses: Delving into the safety of llm-based agents." arXiv preprint arXiv:2311.11855 (2023).

[2] Zeng, Yi, et al. "Air-bench 2024: A safety benchmark based on risk categories from regulations and policies." arXiv preprint arXiv:2407.17436 (2024).

**Questions:**

Please refer to the weaknesses.

---

> ### Author Response · Authors · 2024-11-21
> **Response (1/3)**
>
> Thanks for your thoughtful feedback. Please see our responses in three parts and we will address your concerns accordingly.
>
> > The paper severely lacks a comprehensive literature review and comparison against existing red-teaming methods for agents…
>
> Thank you for providing a pointer to Tian et al. (2023). Upon our close examination of this paper, however, we find it not closely related to the agentic use case our work focuses on: Tian et al. (2023) uses “agents” to focus on LLM roleplaying while we focus on LLMs with access to actual software tools such as APIs or browsers. In a concurrent work AgentHarm [Andriushchenko et al. (2023)], they also point out that it did not test the use of tools and we quote their argument as follows.
>
> > > Tian et al. (2023) investigates harm in multi-agent LLM settings but does not consider tool use or multi-turn interactions.
>
> This area evolves with an exponentially fast pace, which is often a double-sided sword. We are happy to discuss all related work that falls into the category of agent red teaming. We are happy to cite Tian et al. (2023) but would still consider it as bridging work between the base LLM and agentic use, instead of a closely related one to our findings. If there are any other works the reviewer would like to us to cite, please let us know.
>
> In addition to agent red teaming, we are also happy to include more related background knowledge in LLM red teaming and evaluation. We will include the following background information in the Appendix of our revision and point the readers to the Appendix in our main paper.
>
> ### Prompt Jailbreaking
>
> Prompt jailbreaking is the most popular method for jailbreaking and focuses on creating adversarial prompts that bypass model safety and alignment constraints. Chao et al. introduced Prompt Automatic Iterative Refinement (PAIR), an algorithm that generates semantic jailbreaks using an attacker LLM to iteratively refine prompts, achieving high transferability across models like GPT-3.5, GPT-4, and Vicuna (Chao et al., 2023). Their method is able to refine prompts using LLMs as a black box and involves generating an attack and iterating using a judge function to score the results. Jia et al. developed the Improved Greedy Coordinate Gradient (I-GCG), optimizing prompt attacks for diverse templates and achieving higher success rates (Jia et al., 2024). I-GCG builds on GCG, which is an optimization-based method to generate a prompt suffix by maximizing the likelihood of a targeted undesired behavior (Zou et al., 2023). I-GCG uses diverse target templates and multi-coordinate updating strategies, allowing it to bypass safety measures in many LLMs. Lapid et al. explored a genetic algorithm-based approach, optimizing adversarial prompts to disrupt alignment through multi-step reasoning (Lapid et al., 2023). This work uses coordinated prompt crafting to amplify the efficacy of jailbreak attacks.
>
> ### API and System-Level Jailbreaking
>
> System-level jailbreaks target APIs or underlying model mechanisms. Zhang et al. demonstrated Enforced Decoding techniques to directly manipulate generation processes in open-source LLMs, bypassing surface-level alignment (Zhang et al., 2024).
>
> ### Multi-Modal Jailbreaking
>
> Beyond text, multi-modal jailbreaks exploit image and text inputs in models like GPT-4-Vision. Niu et al. proposed a likelihood-based algorithm for discovering image Jailbreaking Prompts (imgJP), which exhibit strong transferability across unseen inputs and demonstrate parallels between text and multi-modal jailbreaks (Niu et al., 2024).
>
> ### Benchmarks and Evaluation
>
> Chao et al. introduced JailbreakBench, an open-source repository with adversarial prompts and a leaderboard to evaluate jailbreak success rates across models (Chao et al., 2024). Zhou et al.’s EasyJailbreak framework also facilitates comprehensive evaluations by modularizing the creation of attack pipelines (Zhou et al., 2024). JAMBench, introduced by Jin et al., evaluates input-output filters in LLM APIs, using encoded prefixes to bypass moderation guardrails (Jin et al., 2024). Zhou et al. proposed EasyJailbreak, a framework combining components from other methods to evaluate prompt-based jailbreaks (Zhou et al., 2024). This work breaks other methods down into four components — Selector, Mutator, Constraint, and Evaluator — to represent the different stages of prompt generation for standardized benchmarking.
>
> ### Encoding-Based Jailbreaking
>
> Encoding methods like ciphers or substitutions present another avenue for attacks. Handa et al. used substitution ciphers to encode adversarial prompts, successfully bypassing GPT-4’s filters (Handa et al., 2023). This method revealed the vulnerabilities in content moderation algorithms relying on direct string matching.

---

> ### Author Response · Authors · 2024-11-21
> **Response (2/3)**
>
> > The benchmark is relatively too small, contains only 100 examples, and from my understanding, is not scalable. Also, the hierarchy of risk categories is not well-supported…
>
> Many safety evaluation benchmarks are relatively small in size as compared to traditional NLP datasets. For example, HarmBench (https://www.harmbench.org) consists of 200 standard behaviors which are expanded from the Trojan Detection Challenge 2023 (TDC) (https://trojandetection.ai/index) that only consists of 100 behaviors. JailBreakBench (https://jailbreakbench.github.io/) consists of 100 prompts, 45% of which are sourced from AdvBench and TDC. A concurrent work, AgentHarm [Andriushchenko et al. (2024)], also consists of only 110 tasks with other being variations of the same task. If we are counting the variations of prompts in our human red teaming data and automated attacks, we can also claim we are releasing a dataset of around 1K prompts; however, we do not wish to confuse the audience between the well-curated behaviors vs. their variations because they have fundamentally different research values.
> On the other hand, as mentioned by the reviewer, benchmarks like Air-Bench-2024 and SorryBench seem to have a lot of prompts. From our past experience with these benchmarks, a major portion of the prompts are paraphrasing and variations of a much smaller set of prompts. This work values quality and diversity much more than quantity. As such, we have spent months just curating for 100 behaviors that cover commonly used websites. Besides just creating chat behaviors, we spent countless efforts on figuring out the right browser behaviors.
>
> Obviously, there is no theoretical issue that would prevent us from creating variations of our well-curated 100 tasks to make it 10x larger. However, we see limited value in doing so in this work because LLM agents are vulnerable to prompts with very obvious malicious intents so paraphrasing or rewrites that intentionally mislead or hide intentions would just make them even worse; this has been validated in our human red teaming experiment. In addition to our base BrowserART benchmark, we also provide human rephrases by expert red-teamers, which serves as a more challenging benchmark on top of our base 100 samples to evaluate browser agents.
> Pointing out safety vulnerabilities is usually not about reporting the average numbers just like capability evaluations for avoiding overfitting and local conclusions. Safety vulnerability is often the worst case result and one single counterfactual example is sufficient to point to the failure of a defense. In our work, what we point out with these 100 well-curated examples is the safety guardrail does not generalize outside the conversational cases, concluding that agent safety is not directly implied from LLM safety (at least for now).
>
> We hope our reply addresses your concern on the dataset size.

---

> ### Author Response · Authors · 2024-11-21
> **Response (3/3)**
>
> > The red-teaming method introduced in the paper lacks novelty. Most examples are manually crafted or transformed from existing examples, and there is no automated way of doing that (or at least not mentioned by the authors). Is it possible to formalize an algorithm to automatically perform red-teaming?
>
> We want to clarify a confusion here. The contributions of this work do NOT include any new red teaming method. All automated methods are existing. We do not aim to propose new algorithms to automatically perform red-teaming. The human red teaming part is done by the authors with tactics already known, so we are merely executing these strategies. At most, we consider ourselves to be showcasing ways to do web agent red teaming but this contribution is relatively minor compared to our empirical results.
>
> This is an evaluation paper, so the major contributions include the toolkit we developed, the behavior dataset, our empirical findings, and the insights from these findings. Our work has important practical impacts to agent developers and model providers. We point out a serious safety vulnerability in agents before autonomous agents can run amok on the Internet.
>
> As highlighted by yourself and Reviewer DmgK, our work is one of the first to uncover this alignment gap. In addition to designing the examples, we evaluate numerous SOTA LLM-agent pairs and show the extremely unsafe behavior of the agents via existing red-teaming approaches. Reviewers DmgK and USQa also highlight the importance of our results in the context of increasing LLM-based agentic use cases. Overall, we believe it is a non-trivial and impactful contribution from our end unrelated to actually introducing red-teaming methods.

---

> > ### Comment · Reviewer_cbCU · 2024-11-21
> > **Increased Score**
> >
> > I would like to thank the authors for their response. Most of my concerns are addressed. I believe this is an appropriate benchmark paper. I have increased my score.

---

> > > ### Author Response · Authors · 2024-11-21
> > >
> > > Thank you for your prompt response and for increasing your review score. We are glad that our response addressed your concerns. We will also update the paper accordingly to give readers a better and more comprehensive understanding.

---

### Official Review · Reviewer_USQa · 2024-11-04

**Soundness:** 4
**Presentation:** 3
**Contribution:** 4
**Rating:** 8
**Confidence:** 3

**Summary:**

The authors curate a dataset to assess the willingness of LLM-based agents to attempt harmful behaviors when interacting with a browser. The authors focus on the substantial within-model difference in behavior between chat- and agent-based deployments. They assess these behaviors using both a text- and an image-based browser framework.

**Strengths:**

Originality: moderate.
Quality: high. Well-chosen, well-scoped hypothesis. Meaningful controls.
Clarity: high. Analyses are well-reasoned and well-presented.
Significance: high. If these results hold, it is a clear call to action to model developers with a demonstrated, focused

**Weaknesses:**

Noticeable grammatical mistakes.

**Questions:**

It would be great to assess the performance of Sonnet 3.5 with computer-use and compare it to the previous version of Sonnet 3.5. Does it seem that Anthropic has identified and/or meaningfully addressed this safety concern in its internal testing?

---

> ### Author Response · Authors · 2024-11-21
> **Response**
>
> Thank you for your encouraging feedback and positive endorsement of our work!
>
>
> The new computer-use (CU) Sonnet-3.5 is indeed an interesting model. Unfortunately Sonnet-3.5-CU was not released when this work was submitted to ICLR. It would make more sense to extend the BrowswerART toolkit to include more applications besides browser behaviors as this model obviously can complete more tasks outside browsers. Also, it would be interesting to see if agent scaffolding combined with Sonnet-3.5-CU is more robust than the rest of LLMs and its previous checkpoint as this CU version is fine-tuned particularly for agentic use. We plan to run this for camera-ready!
>
> Also, for CU models, we believe it would make sense to also develop a GUI safety toolkit. This is a much more difficult task as creating a mirror of a desktop application is more complex than just creating a synthetic website.
>
> We acknowledge the grammatical errors in our manuscript and will review and correct them in our revision.

---

### Official Review · Reviewer_MKMh · 2024-11-04

**Soundness:** 2
**Presentation:** 4
**Contribution:** 2
**Rating:** 6
**Confidence:** 3

**Summary:**

This article introduces BrowserART -- a red teaming framework for evaluating the safety of LLM-based browser agents. The authors contribute a set of 100 harmful behaviors. A red team is expected to convince the evaluated LLMs to perform these harmful behaviors. The article evaluates eight LLMs in several settings. The two main setups are a plain chatbot and a browser agent. The authors show that as chatbots, the LLMs are more resilient to requests for misbehavior. An additional set of conditions includes several commonly acceptable jailbreaking techniques shown here to increase agents' misbehavior dramatically.

**Strengths:**

* Evaluating the misbehavior of LLMs in different conditions is important

* The idea that an LLM may talk and act differently (as humans commonly do) is great!!

* As a red teaming effort, the study achieves remarkable results

**Weaknesses:**

* Substantial and important experimental details are missing
(system prompts are not specified, 2-3 complete specific examples of scenarios provided to both agents and chatbots, details on the tooling of agents).
Examples of prompts are provided in Fig 1. Fig 4, and Appendix B. Only Fig 1, presents a single pair of prompts one of which is expected to fail and the other not. Please provide a few examples from your dataset that any reader can try in a chat.

The system prompt is only mentioned in the context of agents when discussing the context length. The system prompts have determining effect on LLMs' behavior. Therefore all chat and agent system prompts need to be normalized. If using the exact same prompts is not possible then the differences should be disclosed and discussed.

The authors state that they rely on OpenHands for the implementation of the respective agents. However, to make the paper self-contained, some important details related to the browse agent design are missing. For example, which tools are provided to the agent to perform their task? Assuming a full-fledged agent, the tooling should be extensive. Extensive tooling also implies extensive system prompt which may introduce more confounding factors into the agent behavior. Essentially, if the paper claims that it is "morally easier" for an LLM to do a thing (as an agent) rather than to say it (as a chatbot), this claim should be evaluated with the simplest possible agent.
Would an agent equipped with a single tool, e.g., posting a tweet, write in that tweet something that it would not write in a chat? -- of course, when everything else (including the system prompt) remains the same.


* Chatbots and agents are evaluated under different conditions
(differences in queries, different system prompts)

* Following from the previous point, the differences in behaviors cannot be undoubtfully attributed to LLM task (chat vs. agent)

* Clarity of the main goal/purpose of the study.
It is not clear whether the primary purpose of the article is to introduce the BrowserART system or to study the behavior of aligned models?
Which objective is the main one and which is a side track is important to put the right focus on evaluation.

I believe that the attempt to incorporate more contributions by adding the jailbreaking experiments actually harms the paper -- at least if the main focus is comparing agents to chatbots. This significant additional evidence does not contribute to the main question. It also takes precious space and does not allow a deeper comparison of chatbots vs agents. In case both the current title and the jailbreaking experiments remain then the paper must report the behavior of the chatbots under the same jailbreaking conditions as agents.

Alternatively, if the main focus is shifting toward introducing BrowserART and chatbots vs. agents as a side observation, then (1) it requires major rewrites, and (2) comparison to chatbots could naturally be reported only in one of the experiments. Of course, even if the focus is shifted, it is expected that this one experiment would compare chatbots and agents under the same conditions.

**Questions:**

Can the difference in behavior be attributed to different details provided to the agents (to perform their tasks) and not provided to the chatbots?


What are the differences in chat/agent system prompts, and how significant are these differences?


How would the LLM respond to the instructions with actuation details while lacking only the necessary tools (e.g. to access Gmail)?


How do chatbots react to the same attacks attempted on agents?


Is the primary purpose of the article to introduce the BrowserART system or to study the behavior of aligned models?
Which objective is the main one and which is a side track?

---

> ### Author Response · Authors · 2024-11-21
> **Response (1/2)**
>
> Thank you for your feedback! We have addressed your concerns below. Please let us know if our responses address everything or if further clarifications are needed.
>
>
> > Substantial and important experimental details are missing. (system prompts are not specified, 2-3 complete specific examples of scenarios provided to both agents and chatbots…)
>
> Thank you for bringing this to our attention. We used the built-in system prompt provided in the standard frameworks in OpenHands. We will include detailed descriptions of these system prompts in the final draft and provide 2-3 complete specific examples of the scenarios presented to both agents and chatbots to ensure reproducibility of our experiments.
>
> > if the paper claims that it is "morally easier" for an LLM to do a thing (as an agent) rather than to say it (as a chatbot), this claim should be evaluated with the simplest possible agent.
>
> > However, to make the paper self-contained, some important details related to the browse agent design are missing. For example, which tools are provided to the agent to perform their task?
>
> We believe the agent scaffoldings used in this work are straightforward and they are open-sourced so everyone has access to it. OpenHands simply wraps BrowserGym and WebArena agents with other functionalities such as coding and API calling. Our paper only focuses on browser agents so we cite those three works. The underlying agent scaffolding is essentially prompting the LLM with the accessibility tree of the current DOM and allowing it to pick the element to interact with. For SeeAct, it adds bounding boxes to the screenshot of the web page and asks the LLM to pick which bounding box to interact with. The scaffoldings chosen in this work are commonly considered as baseline implementations so we do actually follow the reviewer’s suggestion to choose the simplest possible agents.
>
> We will add more descriptions to the agents we use by utilizing the 10th page. We appreciate the feedback for the unclarity!
>
> > Chatbots and agents are evaluated under different conditions.
>
> **Our experiment is evaluating the robustness of LLM guardrails on out-of-distribution scenarios that are not included in their post-training, namely agentic use in browsers.** As a result, we do expect the evaluations setups are different between the two systems:
> Agents have very long system problems as programmed in their scaffoldings.
> Agents predict which elements to interact with and what interactions to make, while chatbots just respond to texts.
> Agents take prompts + environments (i.e. browser states) as programmed by their scaffoldings while chatbots only take prompts. (In our paper we call prompts “behaviors” since we are asking the LLM to partake in harmful behaviors.)
> We use browser behaviors on agents and chat behaviors on chatbots. The difference is that browser behaviors make specific references to what exists on the screen, such as an existing message recipient (“Tell X they are dumb”), while chat behaviors simply ask for the behavior (“Write a message to tell someone they are dumb”).
>
> For 1 - 3, these deltas are expected because this is part of the experiment setup to measure if the guardrail generalizes to a much more complicated case (i.e. the agent case). For 4,  we explain in Section 3.1 such that in our rephrasing process, we preserve the original semantics of the chat behaviors, making only minimal modifications to ensure the behavior is tailored for a browser agent. We also experimented by using long HTML content from Wikipedia pages as a prefix to all chat behaviors (Section 4.1). We took additional evaluation approaches to ensure that the agents' behaviors did not deviate significantly from the chat behaviors during evaluation.
>
> > Can the difference in behavior be attributed to different details provided to the agents (to perform their tasks) and not provided to the chatbots?
>
> Would you mind clarifying the question a bit? To our understanding, this comment is asking about the cause of the drop of refusals. If this is the case, our response is as follows.
> We hypothesize the cause of the observed behavior difference in Section 5 (Why Agents Are Less Safety-Aligned). We believe current alignment methods do not generalize to browser agent settings. LLMs are only trained on pairs of prompt-response for safety training, which are usually short-context as well. However, in the case of browser agents, the presence of a webpage (simplified HTML) and description of web-related actions make the LLM input distribution different from the input distribution of chatbots. This discrepancy in data distribution makes it possible for the LLM to perform malicious actions as an agent. Moreover, data related to agentic browser use cases might not be well represented in alignment training data, causing further data distribution shift.
>
> However, we might be answering a different question from what you have in mind so we would appreciate your clarification.

---

> ### Author Response · Authors · 2024-11-21
> **Response (2/2)**
>
> > How would the LLM respond to the instructions with actuation details while lacking only the necessary tools (e.g. to access Gmail)?
>
> When the LLM receives instructions that require specific actions involving tools it doesn't have access to (e.g., accessing Gmail), it can inform the user that it cannot perform the requested action due to lack of access. Alternatively, the LLM might hallucinate.
> Note that the browser agents are not given access only to a specific webpage (e.g. Gmail); they can access Chrome and can navigate to any websites they want.
>
> > How do chatbots react to the same attacks attempted on agents?
>
> The attacks we used to red team agents were originally created on chatbots. Evaluation results can be found from existing works such as GCG [Zou et al. 2023] and HarmBench [Mazeika et al. 2023]. Our inline citations of those attacks should provide the reference work to look for the attack results on the LLMs.
>
> > Is the primary purpose of the article to introduce the BrowserART system or to study the behavior of aligned models? Which objective is the main one and which is a side track?
>
> We appreciate the chance to clarify on the primary purpose of our work. This work was motivated by accidental findings when one of the authors attempted to use malicious behaviors on browser agents and expected the agent to reject since the underlying LLMs are refusal-trained. However, that was not the case; the agent actually completed the malicious task and we were shocked. We created BrowserART to systematically research this finding. Our test suite consists of 100 harmful browser behaviors and synthetic websites. Because the original motivation is from the finding where agents built with refusal-trained LLMs do not actually refuse harmful requests, we compare the behavior difference between chatbots and browser agents.
>
> Besides BrowserART, the associated empirical findings and our analysis presented in this paper, our motivation to conduct such experiments is also really significant. Namely, agent safety is not equal to LLM safety and there are a lot of safety vulnerabilities that cannot be examined during the development of LLMs. Building safe agents is not just a responsibility of the LLM provider and agent practitioners have a much better position to build capable and safety-aware agents.
>
> Therefore, the primary objective is to highlight the harmful and unsafe behaviors that refusal-trained LLMs behave differently when used as chatbots and browser agents and call upon model developers to be aware of these issues. To portray this in a systematic way, we create BrowserART, a test suite consisting of harmful browser behaviors and compare the behavior difference between chatbots and browser agents. By publicly releasing this test suite, we aim to raise awareness and provide a valuable tool for researchers to evaluate and improve their LLMs agents.

---

> ### Comment · Reviewer_MKMh · 2024-11-23
>
> Thank you for the clarifications. I will modify the score.
>
> I understand it may be challenging to simplify the agents, and your claim may refer to the entire scaffolding rather than the mere fact of performing harmful actions. The long scaffolding supports the first interpretation of the results in L480. The latter interpretation in L484 could have been supported if the LLM had been provided with a few simple functions (tools) without the large agentic frameworks. Please remove or soften claims that imply that it is "morally easier" for an LLM to do a thing (as an agent) rather than to say it (as a chatbot) because the scaffolding applied by the agentic frameworks is too complex to support such claims.
>
> I am happy to explain and discuss my main concern about the experimental setup exemplified by L199-203.
> When LLMs act as agents, they are given details (such as an email address) that are missing in their prompts when used as chatbots. As such, the behavior change can be attributed to these details rather than to the agentic scaffolding. Such an explanation is unlikely but cannot be disregarded. In other words, the study parameters (long contexts, list of actions, additional details in the prompts, etc.) are not properly isolated in the current experimental setup. This allows awkward interpretations of the results such as attributing the change in behavior to the additional details in the prompts.
>
> Concerning the attacks on chatbots -- the results reported by related work  GCG [Zou et al. 2023] and HarmBench [Mazeika et al. 2023] are not relevant because they used different queries. In order to compare apples to apples, Chatbot w/ attacks should be reported. Then, the reader can see whether the attacks are (equally or more or less) effective against agents as they are effective against chatbots across your scenarios.

---

> > ### Author Response · Authors · 2024-12-03
> >
> > Thank you for your response. We are grateful for the insightful comments and have carefully considered them to improve our manuscript. Please find our detailed responses below.
> > > I understand it may be challenging to simplify the agents, and your claim may refer to the entire scaffolding rather than the mere fact of performing harmful actions. The long scaffolding supports the first interpretation of the results in L480. The latter interpretation in L484 could have been supported if the LLM had been provided with a few simple functions (tools) without the large agentic frameworks. Please remove or soften claims that imply that it is "morally easier" for an LLM to do a thing (as an agent) rather than to say it (as a chatbot) because the scaffolding applied by the agentic frameworks is too complex to support such claims.
> >
> > We understand your concern of fair evaluation and we agree with the experimental setting that you recommend. We will equip chatbots with simple dummy functions, like create_tweet(), and try out experiments for LLMs in chatbot mode. We will update the claims of our second interpretation of our results as per the new results.
> >
> > > I am happy to explain and discuss my main concern about the experimental setup exemplified by L199-203. When LLMs act as agents, they are given details (such as an email address) that are missing in their prompts when used as chatbots. As such, the behavior change can be attributed to these details rather than to the agentic scaffolding. Such an explanation is unlikely but cannot be disregarded. In other words, the study parameters (long contexts, list of actions, additional details in the prompts, etc.) are not properly isolated in the current experimental setup. This allows awkward interpretations of the results such as attributing the change in behavior to the additional details in the prompts.
> >
> > Thank you for raising this concern. We acknowledge that there may be ambiguity in the definition of “details”. The core difference between chatbots and agents is the difference in the input text. For chatbots, the input text is only an instruction, whereas for agents, it is a combination of the action list, webpage representation, and also some informational details like URL, email address, etc. We believe that the difference in inputs to LLMs plays a crucial role which results in the behavior difference. We could have skipped providing the URL/email address details in the initial input prompts to the agent, but this would require human intervention later for the same. Thus, our setup helps us to perform single-turn attacks on the agents as well as reduce human efforts. To address your concern about the potential impact of these details, we plan to include a sanity test where we provide the chatbot with prompts containing similar informational details but without the action list or webpage representation (i.e. the browser behaviors of our benchmark). This will help isolate the effect of these details, and we will include this in our final draft.
> >
> > > Concerning the attacks on chatbots -- the results reported by related work GCG [Zou et al. 2023] and HarmBench [Mazeika et al. 2023] are not relevant because they used different queries. In order to compare apples to apples, Chatbot w/ attacks should be reported. Then, the reader can see whether the attacks are (equally or more or less) effective against agents as they are effective against chatbots across your scenarios.
> >
> > Thank you for your suggestion. Our intention in demonstrating existing adversarial attacks on browser agents was to showcase that even though these attacks are designed for chatbot settings, they are transferable to agentic settings as well. While our focus was not on comparing the degree of effectiveness of the attacks between chatbots and agents, we agree that providing such direct comparisons would be beneficial. We acknowledge this point and consider it a promising direction for future work.

---

### Meta-Review · Area_Chair_BPmL · 2024-12-16

**Metareview:**

The reviewers recommend toning down claims that LLMs find it "morally easier" to perform harmful actions as agents compared to chatbots, as the evidence provided by the complex agentic scaffolding does not fully support this assertion. They also raise concerns about the experimental setup, suggesting that differences in prompts (such as additional details given to agents) could explain behavioral changes, rather than the scaffolding itself. Additionally, they advise comparing attacks on chatbots using the same queries as those used for agents to ensure fairness. The authors have addressed these concerns effectively, and the reviewers now view the paper as a solid benchmark. Additionally, the area chair believes this paper explores an important, emerging, and under-explored area in AI safety: LLM-based browser agents. As a red-teaming effort, the study delivers impressive results and is well-structured throughout. As a result, the Area Chair recommends accepting the paper.

**Additional Comments On Reviewer Discussion:**

The authors have effectively addressed all concerns raised by the reviewers - such as introducing a comprehensive literature review and comparison against existing red-teaming methods.

---

### Decision · Program_Chairs · 2025-01-22

Accept (Poster)